# Flow-DPO: Improving LLM Mathematical Reasoning through Online Multi-Agent Learning

**Yihe Deng**[1,2], **Paul Mineiro**[2]

[1]University of California, Los Angeles
[2]Microsoft Research

## Abstract

Mathematical reasoning is a crucial capability for Large Language Models (LLMs), yet generating detailed and accurate reasoning traces remains a significant challenge. This paper introduces a novel approach to produce high-quality reasoning traces for LLM fine-tuning using online learning **Flows**. Our method employs an incremental output production Flow, where component LLMs collaboratively construct solutions through iterative communication. We train the Flow using online Direct Preference Optimization (DPO) learning with rollouts, generating DPO pairs for each training example and updating models in real-time. We directly compare the quality of reasoning traces generated by our method with those produced through direct model inference, demonstrating the effectiveness of our approach in improving LLM performance in mathematical reasoning tasks.

## 1 Introduction

Mathematical reasoning is a fundamental and vital aspect of Large Language Model (LLM) capabilities, as it is intrinsically linked to logical consistency and problem-solving abilities (Yu et al., 2023; Lu et al., 2023; Zhang et al., 2024c; Gao et al., 2024; Liu et al., 2024). This area has gained significant research interest, partly due to the ease with which results can be verified. Despite the abundance of datasets containing mathematical questions and answers, generating detailed, accurate, and clear reasoning steps remains a significant challenge. While human annotators excel at providing correct answers, their intermediate steps are often too concise or disorganized, rendering the data inadequate for training LLMs. Consequently, researchers increasingly utilize LLM-generated reasoning traces for model fine-tuning. Given the limited feedback provided by mere correctness of final answers, there is growing interest in having the target model generate its own reasoning traces for self-improvement. This approach is particularly relevant in two scenarios: (1) advancing a frontier model (i.e., enhancing a model that is already among the best available), and (2) addressing the high costs associated with using large closed-source models compared to smaller open-source alternatives.

Previous research in this domain has primarily focused on collecting accurate reasoning traces from the model itself through inference and filtering (Zelikman et al., 2022; Yuan et al., 2023; Singh et al., 2023; Hosseini et al., 2024; Pang et al., 2024; Zelikman et al., 2024), subsequently utilizing these traces for Supervised Fine-Tuning (SFT) or Direct Preference Optimization (DPO) (Rafailov et al., 2024). Rejection sampling Fine-Tuning (RFT) (Yuan et al., 2023), a standard and effective approach, augments training data by collecting and filtering unique model responses that yield correct answers. This method is commonly associated with outcome reward, which is based on the final answer. Consequently, another research avenue explores process reward, aiming to generate superior reasoning traces through step-by-step verification or reward mechanisms. While human annotation of each reasoning step has been shown to significantly enhance model performance (Lightman et al., 2023), the substantial cost of such annotations has led researchers to approximate process reward by treating reasoning steps that result in correct answers as preferred steps (Wang et al., 2024a; Zhang et al., 2024b; Lai et al., 2024; Wang et al., 2024b). In essence, given identical training prompts (questions) and desired outcomes (answers), the research community is actively seeking effective and efficient methods to generate high-quality reasoning traces for LLM fine-tuning. This process can be

38th Conference on Neural Information Processing Systems (NeurIPS 2024).

conceptualized as a two-step approach: the data collection step, which aims to identify a "**Better**" operator for trace production, and the SFT step, which "**Compiles**" the collected data into a single LLM model in a System 1 fashion.

This paper focuses on designing a novel and improved pipeline for obtaining high-quality reasoning traces. We directly compare the quality of reasoning traces generated by our method with those produced through direct model inference, using the same volume of data for SFT, filtering on the correct answers and comparing the SFT-ed model performances. Our approach proposes the use of online learning Flows to generate such traces, as opposed to single model inferences. These Flows comprise a collection of component LLMs based on the same architecture, which collaboratively construct solutions through iterative communication (Mineiro, 2024). Specifically, we introduce an incremental productions flow, wherein one LLM generates a limited number of tokens as answer chunks, while another determines whether the maintained partial answer has reached completion. Furthermore, we train our Flow using online DPO learning with rollouts, generating a batch of DPO pairs for each training example at every answer chunk and updating the models as the training data comes in. This core concept aligns with process reward models (PRMs) (Lightman et al., 2023), aiming to generate superior traces incrementally, thus providing denser rewards during fine-tuning. Our method offers greater flexibility by not constraining itself to predefined "reasoning steps". Instead, it allows for adjustable chunk sizes, accommodating fine-grained chunks of mere dozens of tokens and generalizing to outcome reward models when larger chunk sizes are employed. Lastly, our approach remains compatible with further enhancements such as data augmentation and DPO.

## 2  Method

**Incremental Output Production Flow.** We experimented with different flow architectures and achieved the best results with the incremental output production design. As illustrated in Figure 1, this implementation primarily involves two independent LLMs of identical architecture: the Answer LLM and the Stop LLM. The Answer LLM generates one chunk of the response at a time, adhering to a predetermined maximum token limit. We maintain a partial answer, initially empty, to which each newly generated answer chunk is

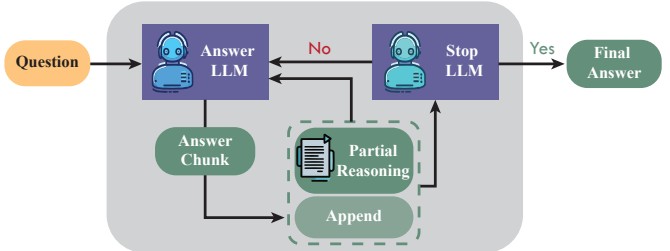

Figure 1: Illustration of the incremental production flow. The Answer LLM is designated to generate an answer chunk with a limited number of tokens. The Stop LLM determines if the current partial answer has reached a satisfying final answer.

appended. This partial answer is then evaluated by the Stop LLM to determine whether the complete response has been achieved. This iterative process continues until the Stop LLM signals the completion of the final answer. Thus, the Flow incrementally constructs the response, with smaller chunk sizes enabling more granular control and larger chunk sizes approximating single-pass model generation. Notably, both the Answer LLM and Stop LLM start from the same base model but are fine-tuned with distinct LoRA adaptors to specialize in their respective tasks.

**Online Flow Learning with Rollouts.** We further enhance the Flow through online DPO learning, incorporating random rollouts at each output node. Figure 2 illustrates this training process. For each input question, the Flow initiates with the Answer LLM generating an answer chunk, continuing until a complete response is produced. Given this output chain, we then perform a random rollout at each output node. For instance, after the initial answer chunk generation and the Stop agent's "No" determination, we allow the Flow to generate an alternative answer chunk, building upon the previous partial answer. This process continues until a second complete answer is reached. If the two answers differ in correctness, we consider them a DPO pair for the Answer LLM, with the chunk leading to the correct answer chosen as the preferred response. Importantly, both the Answer LLM and the Stop LLM are involved in these rollouts and subsequent fine-tuning, with the latter being evaluated on its stopping decisions. For each training instance comprising a question and an answer, we generate a batch of DPO pairs to train both LLMs. This approach enables an online training scheme, updating the models incrementally as new data is processed. This methodology shares similar intuition with the concurrent MCTS-based approaches (Zhang et al., 2024a,b), which

traverses the tree of reasoning steps by selecting the most promising child steps until an answer is reached. From each newly expanded step, they perform a random rollout to estimate the reward of that step. However, we only perform one random rollout at each node without traversing through a tree for better efficiency. Additionally, rather than optimizing over pre-defined reasoning steps, we perform online DPO learning on fine-grained answer chunks.

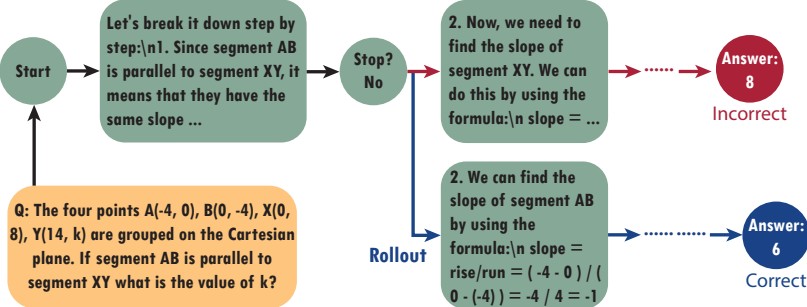

Figure 2: Illustration of the DPO training with rollouts. At each node of the initial generation, we do a random rollout that is different from the original node and continue generation to a final answer. A pair that leads to different answers (correct and incorrect) is considered a DPO training data.

## 3   Results

### 3.1   Experiment Setup.

In our experiments, we consider one LLM model for the entire Flow (Answer LLM and Stop LLM) as well as the Compile step. For the model, we employ two recent and competitive models of different scales: `Llama-3-8B-Instruct` and `Phi-3-medium-128k-instruct` (14B). To investigate the effectiveness of our method, we utilize MetaMath (Yu et al., 2023) as the training dataset. MetaMath is derived from the training data of GSM8K (Cobbe et al., 2021) and MATH (Hendrycks et al., 2021), enhanced through data augmentation techniques. We evaluate the quality of reasoning traces during the Compile step on both GSM8K and MATH datasets. In the Flow learning phase, we use separate LoRA adapters for the Answer LLM and Stop LLM to specialize their capabilities during DPO training. In the Compile phase, we collect an equal amount of data with traces that lead to correct answers from the flow and the baseline, enabling an independent assessment of reasoning quality by examining how it enhances a single model's performance through SFT. We uniformly used a subset of $1,500$ data from MetaMath for all baselines in Compile. For consistency across all baselines, we maintain identical hyperparameters and system prompts in both the SFT process and evaluation.

### 3.2   Progressive Validation Accuracy

We begin by examining the progressive validation accuracy of the Flow during online DPO training with rollouts. Progressive validation accuracy is defined as the cumulative accuracy of the model on incoming training data prior to training:

$$\text{Acc}_{\text{prog}}^{N} = \frac{1}{N} \sum_{i=1}^{N} \mathbb{I}\big(\Theta^{(i-1)}(\mathbf{x}_i) = y_i\big),$$

where $N$ is the number of seen training data, represents the language model fine-tuned on the first $i-1$ data points, $\mathbf{x}_i$ is the $i$-th question in data and $y_i$ is the correct answer. This metric serves as a reliable indicator of the Flow's generalization performance throughout the training process. Figures 3 and 4 illustrate the progressive validation accuracy of our Flow, both with and without training, alongside the zero-shot performance of a single LLM generating reasoning and answers in one step. Without training, the Flow's inference accuracy marginally underperforms that of the standalone model. This discrepancy indicates the Flow's initial inefficiency in managing task-specific requirements, such as explicitly determining when to conclude reasoning or continue based on partial answers. These results highlight the importance of the training process in optimizing the Flow's performance for complex reasoning tasks. Meanwhile, online DPO training effectively enhances the Flow's ability to generalize to new data during online learning across various LLM models. For the `Llama-3-8B-Instruct` model, online DPO learning significantly improves the Flow's performance by $20\%$ within just $2,000$ training instances. Similarly, for the `Phi-3-medium-128k-instruct` model, which demonstrates

strong initial performance in mathematical reasoning with a 79% zero-shot accuracy on the training data, online DPO learning yields a notable improvement of 4 percentage points, reaching nearly 83% accuracy. We note that, the online training scheme enables us to use the progressive validation accuracy as a good indicator for early stopping.

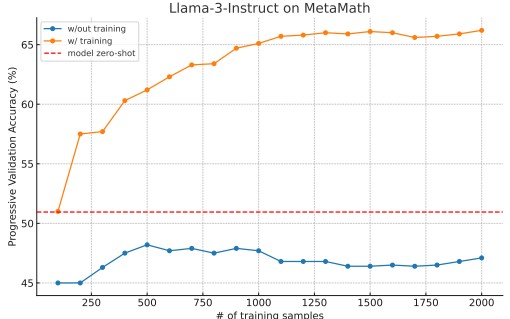 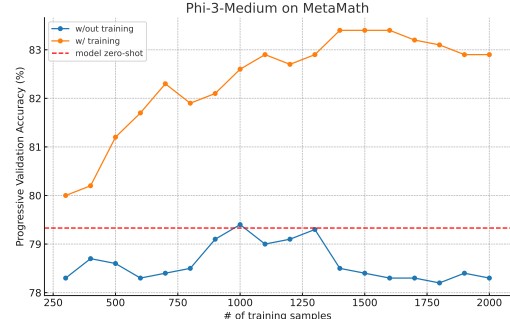

Figure 3: Progressive validation accuracy of Llama-3-Instruct on MetaMath.

Figure 4: Progressive validation accuracy of Phi-3-Medium on MetaMath.

## 3.3 Compile

To assess the quality of flow-generated reasoning traces, we compare them with model-generated traces produced in the Compile step, where we use the collected reasoning traces for SFT on a single LLM. We establish baselines using the model's zero-shot accuracy and its performance after SFT with ground truth traces from the dataset. Additionally, we consider model-generated correct traces for SFT as a strong and widely-used self-training baseline (Yuan et al., 2023; Singh et al., 2023; Hosseini et al., 2024). To ensure a fair comparison of trace quality, we maintain consistent data volumes across all baselines, focusing exclusively on traces that lead to correct answers. The comparative results are presented in Table 1.

Table 1: Main results of comparing the quality of traces used for SFT. We report the accuracy (%) for each model fine-tuned on an identical set of prompts, but with varying answer sources. For Phi-3, we does not include GSM8K due to its already optimized performance on the dataset.

| Model | Method | GSM8K | MATH |
|---|---|---|---|
| Llama-3-Instruct (8B) | 0-shot | 48.9 | 22.3 |
| | SFT (ground-truth) | 67.2 | 25.1 |
| | SFT (self-generated) | 68.8 | 24.2 |
| | **SFT (Flow-generated)** | **71.3** | **27.8** |
| Phi-3-Medium (14B) | 0-shot | - | 35.4 |
| | SFT (ground-truth) | - | 36.3 |
| | SFT (self-generated) | - | 36.5 |
| | **SFT (Flow-generated)** | - | **38.6** |

## 3.4 Qualitative Analysis.

We present a qualitative analysis comparing the reasoning traces generated by our proposed flow method with the ground-truth annotations from the dataset. Through examination of example questions, we demonstrate that while both approaches arrive at correct answers, the flow-generated reasoning traces provide more detailed instructional guidance. To validate these observations, we employed GPT-4o to conduct a systematic qualitative evaluation of response quality. The evaluation results align with our observations, indicating that flow-generated responses (Response B) have better quality. Specifically, it emphasized that flow-generated reasoning traces provides clearer instructional guidance by emphasizing key concepts, such as the negative reciprocal relationship between the slopes of perpendicular lines, while maintaining a logical, step-by-step solution process. The response avoids unnecessary complexity, focusing on essential steps, which improves accessibility and ease of understanding. We provide an additional example in Appendix A.1.

**Question**:
The graph of the equation $x + 2y + 3 = 0$ is perpendicular to the graph of the equation $ax + 2y + 3 = 0$. What is the value of $a$?

**Response A (Ground-truth)**:
The graph of the equation $x + 2y + 3 = 0$ has a slope of $-\frac{1}{2}$. Since the graphs are perpendicular, the slope of the graph of $ax + 2y + 3 = 0$ must be the negative reciprocal of $-\frac{1}{2}$, which is 2. So, the equation $ax + 2y + 3 = 0$ has a slope of 2. To find $a$, we can rewrite the equation as $2y = -ax - 3$ and divide both sides by 2 to get $y = -\frac{a}{2}x - \frac{3}{2}$. Comparing this to the slope-intercept form $y = mx + b$, we see that the slope $m$ is $-\frac{a}{2}$. Since we know the slope is 2, we have $-\frac{a}{2} = 2$. Multiplying both sides by $-2$, we get $a = \boxed{-4}$. The answer is: -4

**Response B (Flow-generated)**:
To solve this problem, we need to understand the concept of perpendicular lines in the context of linear equations. When two lines are perpendicular, their slopes are negative reciprocals of each other.
Let's start by rewriting the given equation $x + 2y + 3 = 0$ in the slope-intercept form, which is $y = -\frac{1}{2}x - \frac{3}{2}$. This tells us that the slope of the line is $-\frac{1}{2}$.
Now, we need to find the slope of the line represented by the equation $ax + 2y + 3 = 0$. We can do this by rewriting the equation in the slope-intercept form, which is $y = -\frac{a}{2}x - \frac{3}{2}$. This tells us that the slope of this line is $-\frac{a}{2}$.
Since the two lines are perpendicular, their slopes are negative reciprocals of each other. This means that $-\frac{1}{2} \cdot (-\frac{a}{2}) = 1$, which simplifies to $a = -4$.
Therefore, the value of $a$ is $\boxed{-4}$.

**GPT-4o Evaluation**:
**Response B** is better, and here's why:

*1. Conceptual Explanation*
**Response A** correctly identifies that the two lines are perpendicular and correctly uses the negative reciprocal relationship between the slopes to solve for $a$. However, it jumps into the algebra without much explanation of why the slopes should be negative reciprocals of each other. **Response B** takes the time to explicitly explain the concept of perpendicularity, emphasizing that the slopes of perpendicular lines are negative reciprocals. This makes the response more educational and ensures the reader understands the reasoning behind the math.

*2. Clarity in Solving*
**Response A** derives the correct equation and solves it correctly but introduces some unnecessary steps. For instance, the intermediate steps involving rewriting the equation as $2y = -ax - 3$ and dividing both sides by 2 are more complicated than necessary. **Response B** simplifies the approach by immediately focusing on the slope-intercept form of both equations. It efficiently explains how to extract the slope and directly compares the slopes of the two equations, making the process more straightforward and easier to follow.

*Conclusion*:
**Response B** provides a clearer and more thorough explanation, focusing on both the concept and the algebra, while avoiding unnecessary complications. Its step-by-step approach makes it more accessible and easier to follow.

# 4 Conclusion

We present an effective approach for generating high-quality reasoning traces using a given LLM, thereby enhancing its mathematical reasoning capabilities. Our method introduces Flow, a multi-agent conversation framework that leverages multiple LLMs to collaboratively solve complex problems through iterative communication. We further refine this framework using online DPO learning with rollouts. Empirical evaluations across various LLM scales on widely-used mathematical benchmarks demonstrate the efficacy of our method, revealing that Flow-generated traces exhibit superior quality compared to both ground-truth and model-generated correct traces. The adaptability of our approach in accommodating different chunk sizes and its applicability to diverse complex reasoning tasks underscore its potential scalability across various scenarios and domains. Future research directions may include optimizing the training process, investigating the impact of increased data, and extending our methodology to other fields requiring sophisticated reasoning capabilities.

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

# A Experiment Details

## A.1 Additional Qualitative Analysis

In the example below, we show an additional qualitative example, which similarly indicates that flow-generated responses (Response B) have better quality. The GPT-4o evaluation emphasized that flow-generated reasoning traces provides structured, step-by-step explanations that enhance comprehensibility. Furthermore, the evaluation highlighted the notable improvement in both clarity and explanatory depth within individual reasoning steps generated by our flow method.

---

**Reasoning Traces Example**

**Question**:
Riku has 25 times more stickers than Kristoff. If Kristoff has x stickers, Riku has 2210 stickers. What is the value of unknown variable x?

**Response A (Ground-truth)**:
To solve this problem, we need to determine the value of x, which represents the number of stickers Kristoff has. We know that Riku has 25 times more stickers than Kristoff. Therefore, Riku has x + 25x = 26x stickers. We also know that Riku has 2210 stickers. We can set up the equation as follows: 26x = 2210 Let's solve for x: x = 2210 / 26 x = 85 The value of x is 85. #### 85 The answer is: 85

**Response B (Flow-generated)**:
Let's break this problem down step by step.
We know that Riku has 25 times more stickers than Kristoff. This means that Riku's sticker count is 25 times Kristoff's sticker count.
We are also given that Riku has 2210 stickers. So, we can set up an equation:
Riku's stickers = 25 × Kristoff's stickers + Kristoff's stickers
Since Riku has 2210 stickers, we can substitute this value into the equation:
2210 = 25 × Kristoff's stickers + Kristoff's stickers
Now, let's simplify the equation by combining the terms:
2210 = 26 × Kristoff's stickers
To solve for Kristoff's stickers, we can divide both sides of the equation by 26:
85

**GPT-4o Evaluation**:
Response B is better, and here's why:

*1. Clarity and Structure*
Response A jumps straight into the math, stating the equation and solving it quickly. However, it lacks the step-by-step breakdown, which is important when explaining the reasoning behind the solution. Response B explains the problem in a more detailed and structured manner. It starts by clarifying the relationship between the number of stickers Riku and Kristoff have and then carefully sets up the equation. This approach is more instructional, making it easier for the reader to follow the logic behind the solution.

*2. Correct Interpretation of the Problem*
Both responses solve the problem correctly by deriving the equation $26x = 2210$ and solving for $x = 85$. Response A has an issue in wording, where it states, "Riku has x + 25x = 26x stickers." While the math is correct, the way it's worded is slightly misleading because saying "25 times more" is interpreted differently than "25 times as many." Response B clarifies this better by stating "Riku's sticker count is 25 times Kristoff's sticker count."

*3. Educational Value*
Response B takes the time to simplify the equation and explain each step, making it more effective for a student or learner to understand the process. It also uses better terminology like "simplify the equation" and breaks down the division, which enhances the educational quality of the response.

*Conclusion*:
Response B provides a more thorough, structured, and pedagogically sound explanation. It guides the user through the problem, making it easier to understand each step.

---

## A.2 Hyperparameters

Experiments were conducted on NVIDIA RTX A100 (80G) GPU clusters. The online DPO fine-tuning process for reasoning trace generation takes approximately 36-48 hours on 4 GPUs. The Compile (SFT) process takes approximately 1 hour on 1 GPU.

Table 2: Online DPO Fine-tuning hyperparameters.

| | |
|---|---|
| Learning rate | 5e-6 |
| Optimizer | `Adam` |
| Global batch size | 32 |
| DPO coefficient $\beta$ | 1.0 |
| Gradient clipping | 1.0 |
| `lora_r` | 8 |
| `lora_alpha` | 8 |
| `lora_dropout` | 0.05 |
| `lora_target` | `all` |
| Maximum steps (chunks) | 6 |
| Chunk size | 160 |

Table 3: Comiple (SFT) hyperparameters.

| | |
|---|---|
| Learning rate | 2e-4 |
| Optimizer | `Adam` |
| Global batch size | 16 |
| Gradient clipping | 1.0 |
| `gradient_accumulation_steps` | 2 |
| `warmup_ratio` | 0.1 |
| `lora_r` | 16 |
| `lora_alpha` | 16 |
| `lora_dropout` | 0.05 |
| `lora_target` | `all` |
| Training epochs | 3 |

## A.3 Prompts

**Prompt for Answer LLM**

<System>
You are a helpful mathematical assistant. Explain your reasoning and then solve the problem.

<User>
{Input Question}

**Prompt for Stop LLM**

<System>
You are an assistant that replies with Yes or No only. In the following task, you are given a Problem and a Candidate Solution. Decide if the Candidate Solution is correct.

<User>
Problem: {problem}

Candidate Solution: {solution}

Is the Candidate Solution correct? Reply with Yes or No only.

**Prompt for GPT-4o Evaluation**

Review the user's question and the corresponding two responses. Determine which response is better.

User: <Question>
Response A: <response A>
Response B: <response B>

After examining the original question, response, and both judgments:
- Explain which response is better and why.
- Conclude with a clear statement of which response is better.

