# OpenReview forum: "Flow-DPO: Improving LLM Mathematical Reasoning through Online Multi-Agent Learning"
_NeurIPS.cc/2024/Workshop/MATH-AI — MATH-AI 24_

### Official Review · Reviewer_6GQK · 2024-09-28
**A reasonable method to improve mathematical reasoning**

**Rating:** 6
**Confidence:** 4

**Review:**

## Summary
Flow-DPO introduces an online multi-agent learning approach to enhance mathematical reasoning in LLMs through incremental output production and random rollouts, generating high-quality reasoning traces.
## Pros
1. Multi-agent collaboration and incremental generation allow for more nuanced decision-making and self-correction, potentially improving reasoning quality.
2. Online learning and random rollouts provide denser, more diverse learning signals, contributing to continuous model performance optimization.
3. The method demonstrates superior performance over traditional self-generated approaches across various LLM scales and multiple mathematical reasoning benchmarks.
## Cons
1. This article lacks an adequate discussion of catastrophic forgetting.
2. While Flow-DPO's design shares similarities with MCTS, the paper does not explore or compare the effectiveness of standard MCTS on similar tasks.
3. Many similar papers have already been proposed.

---

### Official Review · Reviewer_zvxY · 2024-10-03
**General comments**

**Rating:** 6
**Confidence:** 3

**Review:**

**Summary**:
The paper proposes Flow-DPO, a novel multi-agent learning framework using online Direct Preference Optimization (DPO) to generate high-quality reasoning traces for improving Large Language Models (LLMs) in mathematical reasoning tasks. The method demonstrates significant improvements over traditional fine-tuning approaches using self-generated and ground-truth traces on standard mathematical datasets.

**Pros:**
- **Multi-Agent Design**: The use of multiple LLMs (Answer LLM and Stop LLM) to incrementally generate reasoning traces through iterative communication is a novel approach.
- **Use of DPO**: By applying online DPO with rollouts, the paper explores a dynamic and adaptive method for optimizing LLM reasoning, which outperforms standard static preference learning.

**Cons:**
- **Complexity of Training**: The multi-agent training setup, combined with rollouts and online DPO, introduces significant computational overhead, which might limit its applicability to large-scale scenarios.
- **Lack of Qualitative Analysis**: The paper would benefit from more qualitative evaluations, such as visualizing intermediate reasoning steps or comparing different reasoning paths generated by Flow-DPO.

**Questions:**
- How sensitive is the performance to the choice of chunk sizes in the incremental output production flow?
- How does the Stop LLM handle ambiguities or incomplete information in intermediate reasoning steps? Is there a mechanism to revise earlier chunks if later steps conflict with the initial reasoning?

---

### Official Review · Reviewer_7fpD · 2024-10-07

**Rating:** 6
**Confidence:** 4

**Review:**

**Overall**:

This paper presents a novel framework for mathematical reasoning with large language models (LLMs). Instead of relying solely on direct model inference, the authors propose a multi-agent system where the Answer LLM and Stop LLM interact and collaborate across multiple rounds to generate improved reasoning traces. To train the LLMs, the authors adapt the DPO (Direct Policy Optimization) learning method and demonstrate the effectiveness of their approach through extensive simulation studies.

**Strength**:

1. The paper is well-structured with a clear motivation, and the writing and flow are easy to follow.

2. The challenge of handling LLM mathematical reasoning in complex tasks is widely recognized as important, making the problem this paper addresses both timely and significant.

3. The results on the benchmark dataset are promising, demonstrating the advantages of combining the multi-agent framework (Flow) with DPO for training.



**Weakness**:

1. It would be beneficial to provide code or access to the data sources used in the simulation study to facilitate reproducibility.

2. The paper offers limited discussion on computational complexity and hyperparameter tuning (e.g., chunk size, stop criteria), which could be elaborated for better understanding.

3. Further clarification on the novelty of the proposed method compared to existing work in both (1) multi-agent learning and (2) DPO learning with rollouts would help position this contribution more clearly within the literature.



**Minor issue**:
1. Line 116 contains a typo: "represents the language model fine-tuned on the first i1 data points."

---

### Official Review · Reviewer_TVVg · 2024-10-08
**Review of Flow-DPO**

**Rating:** 7
**Confidence:** 3

**Review:**

This paper presents a new method to improve the mathematical reasoning abilities of Large Language Models (LLMs) by producing high-quality reasoning traces through a novel Flow-based approach utilizing Direct Preference Optimization (DPO).  This approach employs two LLMs—one Answer LLM that generates response chunks and a Stop LLM that determines when a full answer has been constructed. This collaboration allows for detailed and step-by-step answer formation.

The paper evaluates the proposed method on two models (Llama-3-instruct and Phi-3-Medium) using MetaMath as the training set, and GSM8K and MATH as the evaluating set and prove the superiority of the proposed method over several baseslines.

---

### Decision · Program_Chairs · 2024-10-09

Accept